# Cryo-EM reveals an unprecedented binding site for NaV1.7 inhibitors enabling rational design of potent hybrid inhibitors

Marc Kschonsak[1†], Christine C Jao[1†], Christopher P Arthur[1], Alexis L Rohou[1], Philippe Bergeron[2], Daniel F Ortwine[2], Steven J McKerrall[2], David H Hackos[3], Lunbin Deng[3], Jun Chen[4], Tianbo Li[4], Peter S Dragovich[2], Matthew Volgraf[2], Matthew R Wright[5], Jian Payandeh[1*], Claudio Ciferri[1*], John C Tellis[2*]

[1]Genentech Inc, Structural Biology, South San Francisco, United States; [2]Genentech Inc, Discovery Chemistry, South San Francisco, United States; [3]Genentech Inc, Neuroscience, South San Francisco, United States; [4]Genentech Inc, Biochemical and Cellular Pharmacology, South San Francisco, United States; [5]Genentech Inc, Drug Metabolism and Pharmacokinetics, South San Francisco, United States

*For correspondence:
jpayandeh@exelixis.com (JP);
ciferri.claudio@gene.com (CC);
tellis.john@gene.com (JCT)

†These authors contributed
equally to this work

Competing interest: See page
13

Reviewing Editor: Jon T Sack,
University of California, Davis,
United States

**Abstract** The voltage-gated sodium (NaV) channel NaV1.7 has been identified as a potential novel analgesic target due to its involvement in human pain syndromes. However, clinically available NaV channel-blocking drugs are not selective among the nine NaV channel subtypes, NaV1.1–NaV1.9. Moreover, the two currently known classes of NaV1.7 subtype-selective inhibitors (aryl- and acylsulfonamides) have undesirable characteristics that may limit their development. To this point understanding of the structure–activity relationships of the acylsulfonamide class of NaV1.7 inhibitors, exemplified by the clinical development candidate **GDC-0310**, has been based solely on a single co-crystal structure of an arylsulfonamide inhibitor bound to voltage-sensing domain 4 (VSD4). To advance inhibitor design targeting the NaV1.7 channel, we pursued high-resolution ligand-bound NaV1.7-VSD4 structures using cryogenic electron microscopy (cryo-EM). Here, we report that **GDC-0310** engages the NaV1.7-VSD4 through an unexpected binding mode orthogonal to the arylsulfonamide inhibitor class binding pose, which identifies a previously unknown ligand binding site in NaV channels. This finding enabled the design of a novel hybrid inhibitor series that bridges the aryl- and acylsulfonamide binding pockets and allows for the generation of molecules with substantially differentiated structures and properties. Overall, our study highlights the power of cryo-EM methods to pursue challenging drug targets using iterative and high-resolution structure-guided inhibitor design. This work also underscores an important role of the membrane bilayer in the optimization of selective NaV channel modulators targeting VSD4.

## Editor's evaluation

This fundamental study describes the structure-based design of novel hybrid inhibitors targeting a human sodium channel which is a pain target. Exceptionally strong evidence for key claims was produced with a structural biological pipeline for iterative structural determination of drugs complexed with an engineered sodium channel. This work is expected to be of interest to biophysicists, drug developers, neurobiologists and pain researchers.

## Introduction

Voltage-gated sodium ($Na_V$) channels initiate and propagate action potentials in excitable cells and play important roles in health and disease (*Catterall et al., 2005*; *Ahern et al., 2016*). The $Na_V$1.7 channel is expressed predominantly in the peripheral nervous system, and genetic studies have identified compelling loss-of-function and gain-of-function phenotypes in human pain syndromes, prompting significant efforts to develop $Na_V$1.7-selective inhibitors as potential novel analgesic drugs (*Dib-Hajj et al., 2013*; *Payandeh and Hackos, 2018*; *McKerrall and Sutherlin, 2018*). $Na_V$ channels contain 24-transmembrane segments linked in four homologous domains (DI-DIV), where four peripheral voltage-sensor domains (VSD1-4) surround a central ion-conducting pore module that houses the ion selectivity filter and key ligand and toxin binding sites. Traditionally, all clinically available $Na_V$ channel inhibitors lack significant molecular selectivity among the $Na_V$1.1–1.9 subtypes owing to the high sequence conservation found at the ligand binding site within the central cavity of the ion-conducting pore module (*McKerrall and Sutherlin, 2018*; *de Lera Ruiz and Kraus, 2015*).

A breakthrough study in 2013 by McCormack and colleagues reported the discovery of an arylsulfonamide antagonist, **PF-04856264**, that bound to an unprecedented receptor site in VSD4 with demonstrated molecular selectivity for human $Na_V$1.7 over other subtypes (*McCormack et al., 2013*). However, the related development candidate **PF-05089771** did not meet clinical endpoints in human subjects with painful diabetic peripheral neuropathy, possibly due to poor target coverage and the intolerable doses required to achieve efficacy (*Swain et al., 2017*; *McDonnell et al., 2018*; *Siebenga et al., 2020*). Additionally, an alternate acylsulfonamide inhibitor that also targeted the VSD4 receptor site in $Na_V$1.7, **GDC-0276**, was halted in phase I clinical trials due to safety concerns and potential off-target effects likely attributed to the high lipophilicity of the compound (*Rothenberg et al., 2019*; *Safina et al., 2021*). To date, selective $Na_V$1.7 inhibitors with an improved therapeutic index relative to these clinical-stage compounds have not yet been identified, and this absence underscores the need to optimize such molecules using modern, structure-guided design approaches.

The inherent complexity and dynamic nature of human $Na_V$ channels have historically presented significant barriers to obtaining high-resolution experimental structural information, especially of inhibitor-bound complexes (*Noreng et al., 2021*). Using an engineered human VSD4 $Na_V$1.7-$Na_V$Ab bacterial channel chimera and X-ray crystallography, the binding mode of the **GX-936** arylsulfonamide inhibitor revealed that the anionic sulfonamide group engages the fourth arginine gating charge (R4) to trap VSD4 in an activated conformation, which in turn stabilizes a non-conductive, inactivated state of the channel (*Ahuja et al., 2015*). While determinants of subtype selectivity and structure–activity relationships (SAR) of the arylsulfonamide inhibitor class could be rationalized by the **GX-936** co-crystal structure, additional ligand-bound structures of suitable resolution were not returned by the $Na_V$1.7-$Na_V$Ab chimeric channel crystallography system. This shortcoming led molecular docking studies to presume that the acylsulfonamide inhibitors also bound VSD4 in an analogous manner, despite points of inexplicable SAR (*Sun et al., 2019*). The overall paucity of direct structural information has made the optimization of $Na_V$1.7 inhibitors challenging, with many questions about the determinants of potency, selectivity, and the relationship between the acylsulfonamide and arylsulfonamide inhibitor classes remaining unanswered.

Over the last three decades, protein crystallography and structure-based drug design (SBDD) have become gold standards across the pharmaceutical industry for the identification of ligand binding pockets and the optimization of drug candidates for clinical development. While SBDD has proven successful for many important targets, including G protein-coupled receptors (GPCRs) (*Congreve et al., 2014*), its applicability to several membrane protein targets, such as $Na_V$ channels, has been limited due to the extreme difficulties with their iterative crystallization and structure determination (*Noreng et al., 2021*). Cryogenic electron microscopy (cryo-EM) has recently emerged as a transformative technique to determine the high-resolution structures of diverse protein targets and has proven particularly effective for membrane proteins that are recalcitrant to crystallization (*Cheng, 2018*). Despite this breakthrough, cryo-EM structure determination of membrane targets in complex with small or large molecule therapeutics frequently remains retrospective and is often enabled after the advancement of key molecules into the clinic (*Balestrini et al., 2021*; *Rougé et al., 2020*). Slow turnaround times and modest resolutions typically offered by cryo-EM have limited its application for real-time SBDD efforts. Here, we describe a system for the iterative determination of high-resolution $Na_V$1.7-VSD4 small molecule co-structures via cryo-EM that has led to the development of a novel

class of inhibitors. Critical to this advancement has been the establishment of a robust protocol for sample preparation and structure determination and the first structure of an acylsulfonamide bound to Na$_V$1.7-VSD4, revealing a previously unknown inhibitor binding pocket between the S3 and S4 helices. Consequently, a novel hybrid inhibitor series bridging the aryl and acylsulfonamide pockets was designed and validated. Our work exemplifies the deployment of cryo-EM as a workhorse structural biology tool in an active medicinal chemistry campaign and thus represents an important milestone toward Na$_V$ channel drug discovery.

## Results

### Structure of the GNE-3565-VSD4 Na$_V$1.7-Na$_V$Pas channel complex in lipid nanodiscs

Our approach to establish an iterative, high-resolution system to enable Na$_V$1.7 SBDD was ultimately guided by three observations: our repeated failure to generate well-diffracting crystals of the Na$_V$1.7-Na$_V$Ab bacterial channel chimera system (*Ahuja et al., 2015*), our difficulty of reproducibly expressing a suitable amount of full-length human Na$_V$1.7 channel (*Xu et al., 2019*; *Clairfeuille, 2019*), and the very limited local resolution observed at VSD4 in all available human Na$_V$1.7 cryo-EM structures (*Shen et al., 2019*; *Huang et al., 2022*). Thus, we sought to exploit an engineered human VSD4-Na$_V$1.7-Na$_V$Pas cockroach channel chimeric construct that had been previously shown to complex with small molecule inhibitors and peptide toxins known to target VSD4 (*Clairfeuille, 2019*). The recovered protein yields following expression and purification (50 µg/L) allowed us to readily pursue reconstitution of the VSD4-Na$_V$1.7-Na$_V$Pas channel into lipid nanodiscs (*Figure 1*, *Figure 1—figure supplement 1*). Small molecule inhibitors were added prior to sample vitrification, followed by cryo-EM data collection and processing procedures, which allowed us to routinely obtain 3D-reconstructions in the 2.2–3.0Å resolution range around the VSD4 inhibitor binding site (*Figure 1*).

The VSD4-Na$_V$1.7-Na$_V$Pas chimera displays the expected domain-swapped arrangement with numerous densities assigned as phospholipids bound to the channel, confirming the maintenance of a native membrane-like environment (*Figure 1*). As seen in previous Na$_V$Pas structures (*Clairfeuille, 2019*; *Shen et al., 2017*; *Shen et al., 2018*), the ion-conducting pore module of the VSD4-Na$_V$1.7-Na$_V$Pas chimeric channel is closed, consistent with a nonconductive or inactivated state. The quality and resolution of our structure allowed us to assign 111 water molecules for the highest resolution structure (*Figure 1E*, *Video 1*). The visualization of many well-resolved water molecules bound within the VSDs and coordinated within the ion selectivity filter provides new insights into the interactions that might occur during gating charge transfer and ion conduction, respectively.

Although we were able to determine the crystal structure of VSD4-Na$_V$Ab bound to the arylsulfonamide inhibitor **GX-936** (*Ahuja et al., 2015*), no well-diffracting crystals were obtained for any other arylsulfonamide compounds despite years of continued effort. Arylsulfonamide **GNE-3565** is representative of an advanced series of arylsulfonamide class Na$_V$1.7 inhibitors that demonstrates channel blockage at subnanomolar concentrations with mixed subtype selectivity (*Figure 2A*). We complexed **GNE-3565** with the VSD4 Na$_V$1.7-Na$_V$Pas channel-nanodisc system and employed cryo-EM to assess its overall binding pose (*Figure 2B and C*). The 2.9 Å resolution cryo-EM map revealed that the S1-S2 and S3-S4 helices from VSD4 form a clamshell-like structure that closes over **GNE-3565** (*Figure 2D and E*, *Figure 2—figure supplement 1*, *Figure 2—figure supplement 2*, *Supplementary file 1A*), similar to the binding pose reported for **GX-936** (*Ahuja et al., 2015*). Specifically, the ionized arylsulfonamide group of **GNE-3565** (measured pKa = 5.8) bisects VSD4 to salt-bridge directly to R4 on the S4 helix, while the central phenyl ring extends perpendicular between the S2 and S3 helices to directly contact established subtype-selectivity determinants Tyr1537 and Trp1538 on the S2 helix (*Figure 2A–D*). As for **GX-936**, the **GNE-3565,** arylsulfonamide VSD4 receptor binding site can be divided into three regions: an anion-binding pocket, a selectivity pocket, and a lipid-exposed pocket (*Figure 2D*). The root-mean-square deviation (RMSD) between the **GNE-3565**-VSD4 and the **GX-936**-VSD4 structures (*Ahuja et al., 2015*) is 0.667 (714–714 atoms) (*Figure 2—figure supplement 2*), supporting long-held assumptions by medicinal chemistry teams that all structurally related arylsulfonamides should be expected to complex to VSD4 through similar determinants (*Swain et al., 2017*; *Weiss et al., 2017*; *Graceffa et al., 2017*; *McKerrall et al., 2019*; *Roecker et al., 2021*; *Focken et al., 2016*; *Wu et al., 2017*).

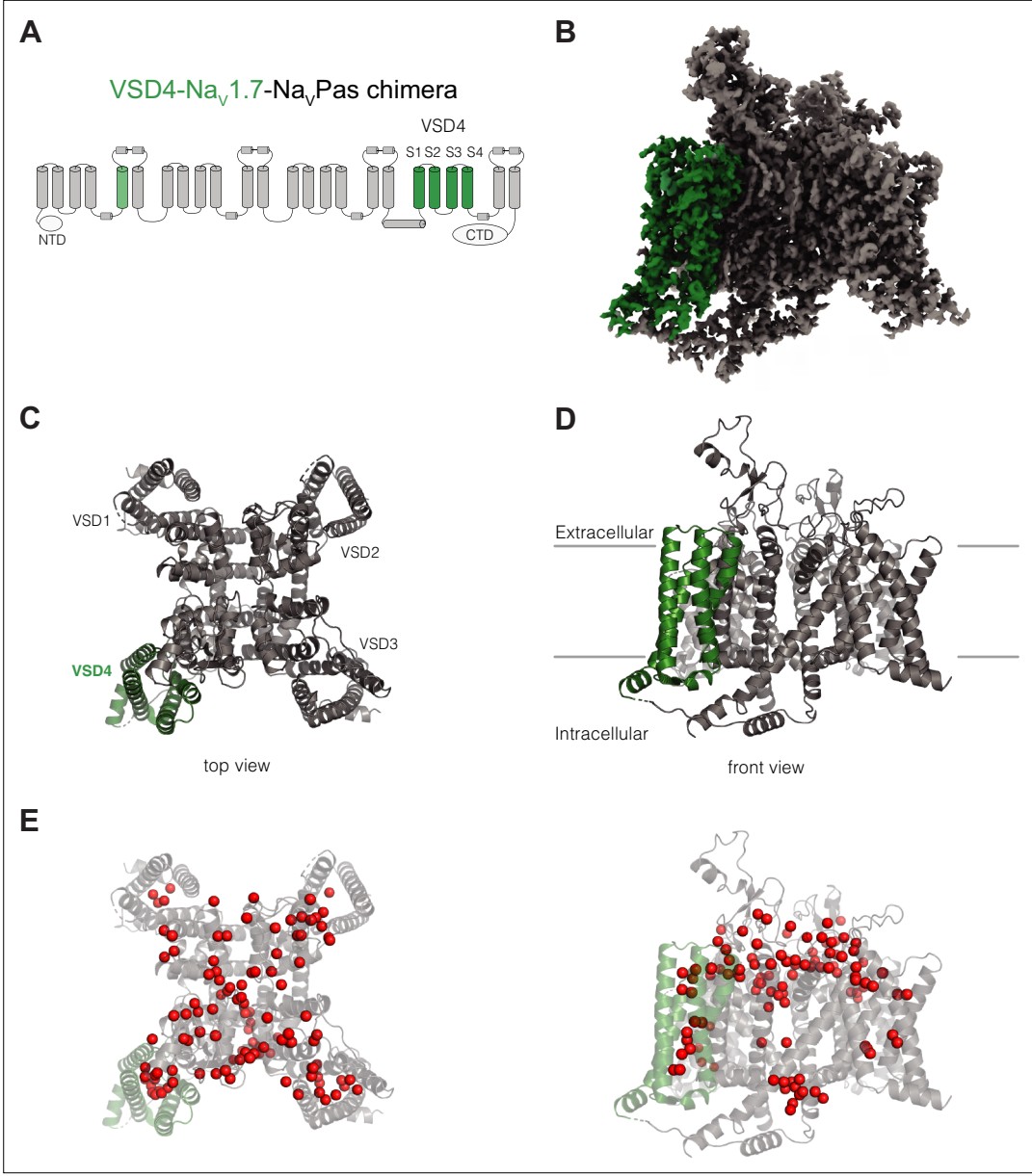

**Figure 1.** Structure of VSD4-Na$_V$1.7-Na$_V$Pas. (**A**) Schematic of the VSD4-Na$_V$1.7-Na$_V$Pas channel. The portions humanized to the Na$_V$1.7 sequence are shown in green. N-terminal domain (NTD) and CTD are indicated. (**B**) Side view of the single-particle cryogenic electron microscopy (cryo-EM) reconstruction of VSD4-Na$_V$1.7-Na$_V$Pas channel. (**C, D**) Cartoon representations of the top and side views of VSD4-Na$_V$1.7-Na$_V$Pas channel. Individual VSD domains are indicated. VSD4 is highlighted in green. (**E**) Localization of water molecules (in red) in the VSD4-Na$_V$1.7-Na$_V$Pas channel structure. VSD4 is highlighted in green.

The online version of this article includes the following source data and figure supplement(s) for figure 1:

**Figure supplement 1.** Purification of VSD4-Na$_V$1.7-Na$_V$Pas channel.

**Figure supplement 1—source data 1.** Source data of VSD4-Na$_V$1.7-Na$_V$Pas purification.

## Structure of the acylsulfonamide GDC-0310 in complex with the VSD4 Na$_V$1.7-Na$_V$Pas channel

To investigate the binding pose of a representative Na$_V$1.7-selective acylsulfonamide, **GDC-0310** (*Figure 3A*) was complexed with the VSD4 Na$_V$1.7-Na$_V$Pas channel in nanodiscs and a cryo-EM structure was determined to 2.5 Å resolution (*Figure 3B–D*, *Figure 3—figure supplement 1*, *Supplementary file 1A*). Remarkably, the binding mode of **GDC-0310** revealed an unexpected and previously

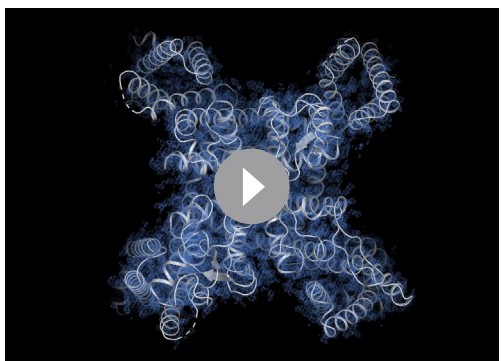

**Video 1.** Movie illustrating the quality of VSD4-Na$_V$1.7-Na$_V$Pas channel bound to GNE-1305. Several water molecules can be observed.

https://elifesciences.org/articles/84151/figures#video1

unknown pocket that formed between the S3 and S4 helices (*Figure 3B–E*). While the aryl and acyl moieties of **GDC-0310** and **GNE-3565** bind in the same region, the remainder of the acyl receptor site is orthogonal to the corresponding binding domain reported for the aryl class (*Figure 3B–D*). In detail, the anionic acylsulfonamide of **GDC-0310** participates in salt-bridge bonding interactions with R4 and R3 through the carbonyl and sulfonamide oxygen atoms, respectively (*Figure 3F*). Moreover, the aryl ring of the **GDC-0310** splits the S3 and S4 helices to occupy a lipophilic pocket, displacing the S3 helix laterally by ~3 Å relative to the **GNE-3565**-VSD4 structure (*Figure 3G*). Our cryo-EM structures serve to highlight a dynamic environment in VSD4 where the S1-S4 helices can be differentially separated by the binding of distinct small molecule inhibitors (*Figure 2D* and *Figure 3D*).

It is notable that the cyclopropyl substituent of **GDC-0310** makes effective van der Waals interactions with I1574 and I1588 from S3 and I1601 and I1604 on S4 (*Figure 3G*), while the alpha-methylbenzylamine 'tail' region sits almost entirely within the lipid bilayer (*Figure 3A and C*), where

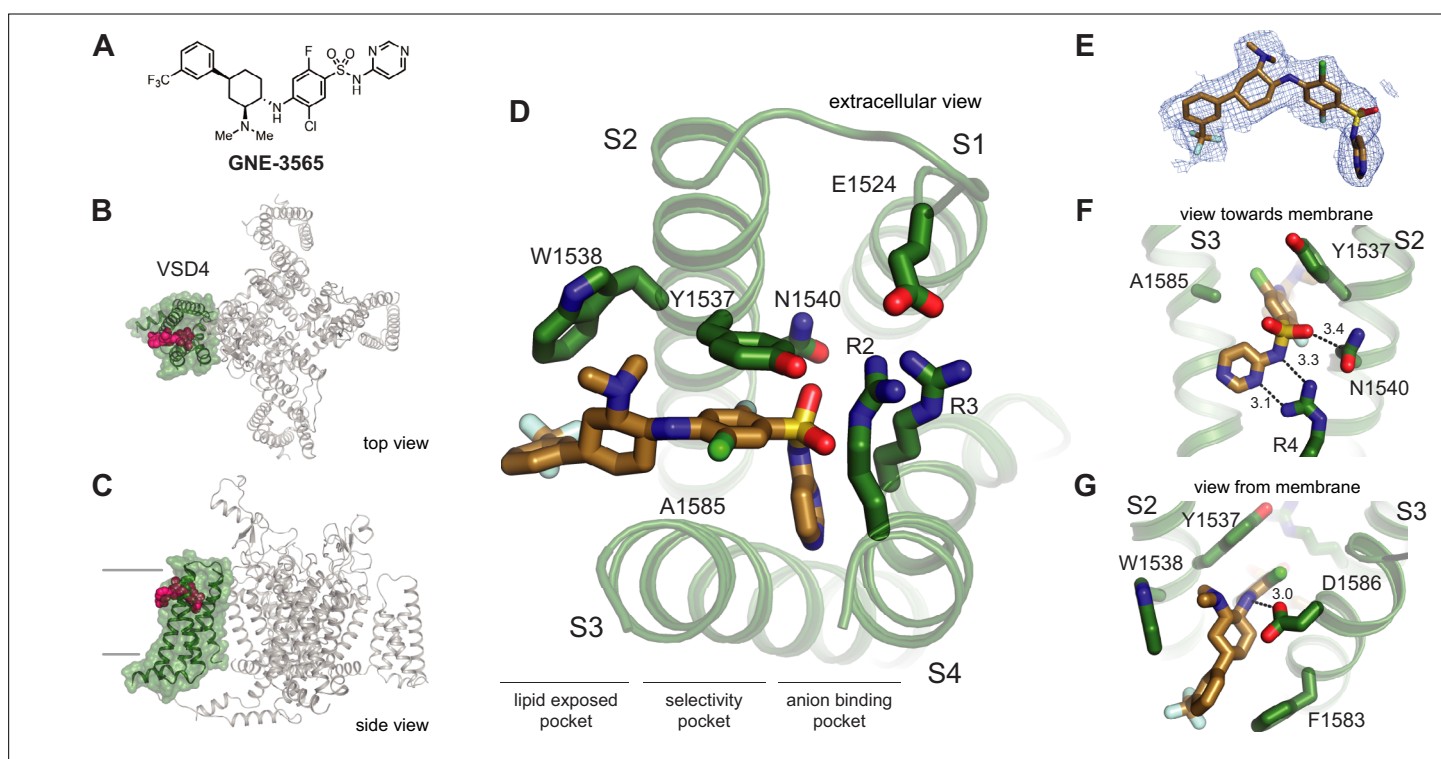

**Figure 2.** Structure of GNE-3565 bound to VSD4-Na$_V$1.7-Na$_V$Pas. (**A**) Chemical structure of arylsulfonamide GNE-3565. (**B, C**) Top and side views of VSD4-Na$_V$1.7-Na$_V$Pas channel bound to **GNE-3565**. VSD4 is highlighted in green, **GNE-3565** in magenta. (**D**) Extracellular view of VSD4-Na$_V$1.7-Na$_V$Pas arylsulfonamide receptor site is shown with select side chains rendered as sticks. (**E**) The cryogenic electron microscopy (cryo-EM) map surrounding the ligand **GNE-3565** is shown in mesh representation. (**F**) View toward the membrane highlighting key interactions with the **GNE-3565** anionic group. (**G**) View from the membrane highlighting key interactions with the **GNE-3565** central phenyl ring.

The online version of this article includes the following figure supplement(s) for figure 2:

**Figure supplement 1.** Cryo-EM processing workflow of GNE-3565 and GNE-9296.

**Figure supplement 2.** Comparison of VSD4-Na$_V$1.7 bound to the arylsulfonamides GNE-3565 and GX-936 (PDB:5EK0).

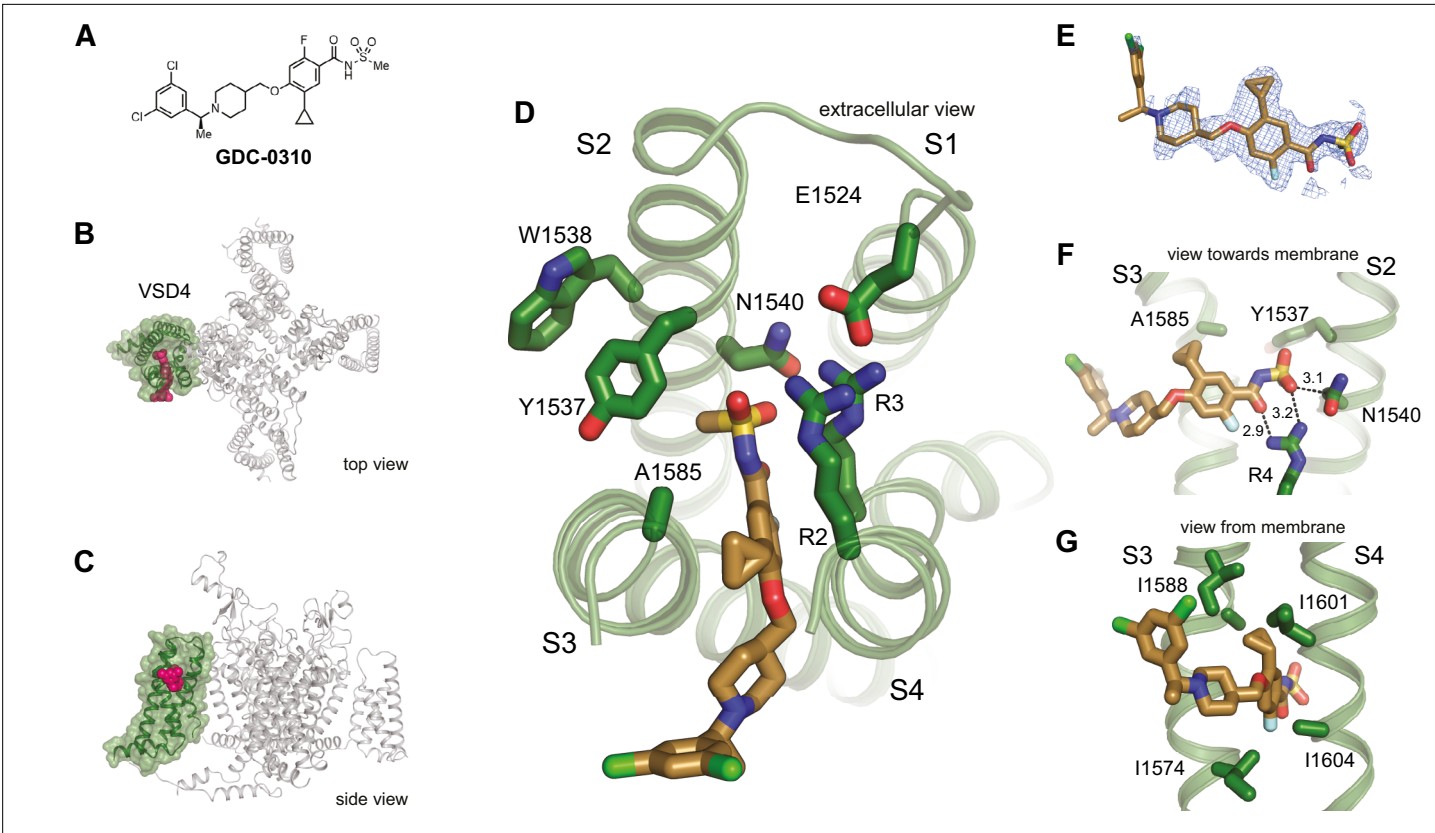

**Figure 3.** Structure of **GDC-0310** bound to VSD4-Na$_V$1.7-Na$_V$Pas. (**A**) Chemical structure of acylsulfonamide **GDC-0310**. (**B**) Top and side views of VSD4-Na$_V$1.7-Na$_V$Pas channel bound to **GDC-0310**. VSD4 is highlighted in green, **GDC-0310** in magenta. (**D**) Extracellular view of VSD4-Na$_V$1.7-Na$_V$Pas acylsulfonamide receptor site is shown with select side chains rendered as sticks. (**E**) The cryogenic electron microscopy (cryo-EM) map surrounding the ligand **GDC-0310** is shown in mesh representation. (**F**) View toward the membrane highlighting key interactions with the **GDC-0310** warhead. (**G**) View from the membrane highlighting van der Waals interactions with the **GDC-0310** cyclopropyl substituent. The alpha-methylbenzylamine tail region sits almost entirely within the lipid bilayer.

The online version of this article includes the following figure supplement(s) for figure 3:

**Figure supplement 1.** CyroEM processing workflow for GNE-1305 and GDC-0310.

density for this portion of the inhibitor is only poorly resolved (*Figure 3E*). Considering the relative depth of the **GDC-0310** binding site in relation to the membrane-water interface (~8 Å), our structure suggests a membrane access pathway for acylsulfonamides to the VSD4 receptor site, which has important implications for understanding the pharmacology, SAR, and potential development liabilities of the inhibitor series (*Payandeh and Volgraf, 2021*). Notably, this structure also reveals that the Tyr1537 side chain on S2 exists in a down rotamer conformation and does not contact **GDC-0310** directly, which provides the first direct structural rationale for why the selectivity profiles between the acyl and arylsulfonamide classes differ substantially (*Figure 3D*, *vide infra*; *Bankar et al., 2018*).

## Structure-based rational design of hybrid Na$_V$1.7 inhibitors

Upon inspection of the superimposed **GNE-3565**-aryl and **GDC-0310**-acyl structures, we could immediately envision a novel class of hybrid inhibitors that would simultaneously occupy both the aryl- and acylsulfonamide binding pockets (*Figure 4A and B*). Such an enlarged hybrid binding pocket would offer unique opportunities to gain potency directly from ligand–protein interactions, which might enable removal of the hydrophobic lipid-facing tail groups seen in all previous generations of VSD4-targeting Na$_V$ channel inhibitors. These groups have typically been critical for maintaining potency, but use of the plasma membrane to drive membrane-bound target occupancy is largely nonspecific, and potentially introduces off-target liabilities that can contribute to toxicity. However, it was unclear whether this hybrid conformation of the VSD4 domain could accommodate

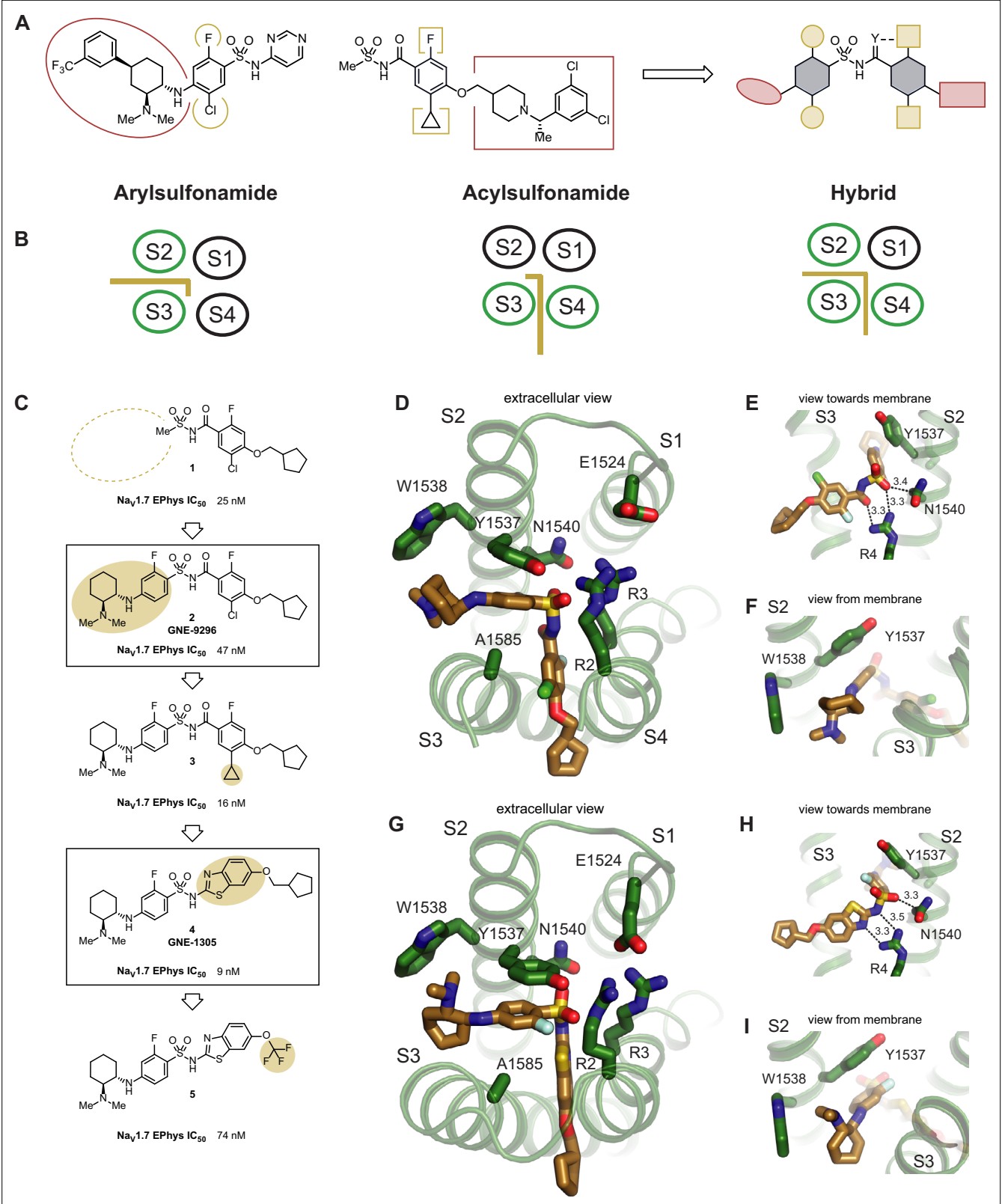

**Figure 4.** Structure-based design of potent hybrid inhibitors of Na$_V$1.7. (**A**) Illustration of hybrid molecule design approach. (**B**) Arylsulfonamide, acylsulfonamide, and hybrid molecule poses. (**C**) Hybridization strategy and molecule optimization. Highlighted are molecule **2** (GNE-9296) and molecule **4** (GNE-1305). (**D**) Extracellular view of VSD4-Na$_V$1.7-Na$_V$Pas bound to the hybrid molecule **2** (GNE-9296). (**E**) View toward the membrane highlighting key interactions with the anionic group. (**F**) View from the membrane highlighting the lack of a stacking interaction between Y1537 and the

*Figure 4 continued on next page*

*Figure 4 continued*

phenyl ring. (**G**) Extracellular view of VSD4-Na$_V$1.7-Na$_V$Pas bound to the hybrid molecule **4** (GNE-1305). (**H**) View toward the membrane highlighting key interactions with the anionic group. (**I**) View from the membrane highlighting the p-stacking interaction between Y1537 and the phenyl ring.

The online version of this article includes the following figure supplement(s) for figure 4:

**Figure supplement 1.** Biophysical and pharmacological characterization of human Na$_V$1.7 channel by using Syncropatch384.

**Figure supplement 2.** Density maps of GNE-9296 and GNE-1305.

**Figure supplement 3.** Multiple sequence alignment of human VSD4 Na$_V$1.

the described inhibitors and/or whether it was accessible during any stage of Na$_V$1.7 channel gating. We therefore set out to demonstrate a proof of concept that appropriately designed small molecules could induce this hypothesized binding site while inhibiting channel function with meaningful potency.

**GDC-0310** was a relatively unattractive starting point for hybridization because of its high molecular weight (543 g/mol) and lipophilicity (cLogP = 5.2). Accordingly, re-evaluation of potency and physicochemical property data available for previously synthesized Na$_V$1.7 inhibitors identified progenitor compound **1**, which offers reasonable potency against Na$_V$1.7 (IC$_{50}$ = 25 nM, n = 8; *Figure 4C*, *Figure 4—figure supplement 1*) and reduced molecular weight (350 g/mol) and lipophilicity (cLogP = 2.7). Seeking to leverage existing knowledge of the arylsulfonamide binding pocket, **GNE-9296** (compound **2**) was synthesized in an attempt to directly graft on a significant portion of the **GNE-3565** scaffold. Gratifyingly, this molecule retained substantial potency, affording channel blockade at an IC$_{50}$ of 47 nM (n = 8).

A cryo-EM structure of **GNE-9296** bound to the VSD4 Na$_V$1.7-Na$_V$Pas channel confirmed that the molecule adopts the desired hybrid binding mode (*Figure 4D*, *Figure 4—figure supplement 2A*). Here, the anionic group of the novel hybrid compound binds in the same position as in the aryl and acyl poses, interacting closely with R4, as seen for the aryl-like binding (*Figure 4E*). The aryl- and acylsulfonamide-derived structural motifs of **GNE-9296** occupy their expected positions in the pockets formed between S2/S3 and S3/S4, respectively. Notably, in this hybrid configuration, Y1537 adopts a similar arrangement as observed for the arylsulfonamides (*Figure 4F*).

Encouraged by this result, we hoped to draw on previously established knowledge of acylsulfonamide SAR to improve the potency of this compound. Established SAR within the acylsulfonamide series suggested that replacement of the chloro substituent on the benzamide fragment of the molecule with a cyclopropane would typically improve potency. This SAR proved translatable to the hybrid class molecules, resulting in inhibitor **3** (IC$_{50}$ = 16nM, n = 12; *Figure 4C*, *Figure 4—figure supplement 1*, *Figure 4—figure supplement 2A*).

Further analysis of the cryo-EM structure of **GNE-9296** revealed that the aryl ring in the S2/S3 binding pocket was shifted ~1 Å away from the VSD4 core compared to arylsulfonamides such as **GNE-3565** (data not shown). The aryl ring in the S3/S4 pocket exhibited better overlap with the analogous substituents in the **GDC-0310** structure. We hypothesized that the acylsulfonamide moiety of **GNE-9296**, which places the two aryl substituents 4.7 Å apart, might not be fully optimized for use in hybrid inhibitors. In comparison, an arylsulfonamide moiety would position these two groups at a distance of ~3.3 Å. On this basis, we designed *N*-benzothiazolyl sulfonamide **GNE-1305** (compound **4**), which demonstrated potent Na$_V$1.7 inhibition (IC$_{50}$ = 9nM, n = 8, *Figure 4C*, *Figure 4—figure supplement 1*, *Figure 4—figure supplement 2B*).

Our last hypothesis was again validated with a cryo-EM co-structure of **GNE-1305** bound to the VSD4 Na$_V$1.7-Na$_V$Pas channel (*Figure 4G*, *Figure 4—figure supplement 2B*). Here, we observed the typical face-on pi-stacking orientation of the S2/S3 substituent with Y1537 while maintaining efficient space filling in the S3/S4 pocket. Moreover, the efficient occupation of the non-membrane exposed regions of the binding pocket permitted the removal of the lipophilic tail of the molecule in **5**, where a CF$_3$ group replaces the cyclopentylmethyl substituent. Although this molecule shows reduced potency (IC$_{50}$ = 74 nM, n = 8), lack of substantial small molecule contact with the lipid bilayer is an uncommon feature in all other known Na$_V$1.7 VSD4 domain inhibitors. These molecules serve as a key proof of concept, highlighting the power of cryo-EM to yield new structural insights that can inspire the design of novel, structurally differentiated small molecule Na$_V$ channel inhibitors.

# Discussion

Iterative structure-based design is among the most powerful techniques to facilitate the development of small molecule drug candidates. Although a variety of technologies are capable of providing information about small molecule ligand–protein interactions, SBDD has traditionally been synonymous with the use of X-ray crystallography, with other techniques failing to rival its high resolution and fast cycle times. Here, we have demonstrated that a robust cryo-EM structure pipeline is capable of quickly providing co-structures at resolutions suitable for use in an active medicinal chemistry optimization campaign. Moreover, we have highlighted the power of structural methods, and cryo-EM in particular, to provide new and important insights into inhibitor design through the discovery of a previously unknown binding mode for acylsulfonamide $Na_V1.7$ inhibitors. We further note that our use of an engineered VSD4-$Na_V1.7$-$Na_V$Pas chimeric channel construct was essential to produce sufficient yields of recombinant protein capable of routinely returning multiple cryo-EM samples per preparation. This system repeatedly delivered high-resolution depictions of the VSD4 inhibitor binding site and thus contrasts with prior accounts of VSD4 in cryo-EM structures of full-length human $Na_V1.7$ channels (*Shen et al., 2019*; *Huang et al., 2022*) Importantly, high-quality cryo-EM maps of VSD4 in complex with ICA121431 have recently been reported from full-length human $Na_V1.3$ channel protein (*Li et al., 2022*), indicating that SBDD efforts targeting a particular $Na_V$ channel subtype would likely benefit from exploring both engineered and native channel sample options at early project stages.

The co-structure of **GDC-0310** was critical to contextualize historical differences between aryl- and acylsulfonamide $Na_V1.7$ inhibitors, which include divergent SAR in the linker phenyl ring and tail regions, disparate $Na_V$ family selectivity patterns, and differences in on-rates and off-rates (*Safina et al., 2021*; *Sun et al., 2019*; *Bankar et al., 2018*; *Luo et al., 2019*; *DiMauro et al., 2016*; *Focken et al., 2018*). Previous efforts to model acylsulfonamide binding have attempted to dock molecules into the **GX-936** co-structure and have proposed subtle changes in molecular register and/or pose to explain the non-translatable SAR (*Sun et al., 2019*; *Kotla et al., 2019*). Here, we demonstrate that the molecular features of aryl- and acylsulfonamide inhibitors diverge because they occupy distinct, but overlapping, pockets within the VSD4 domain.

The selectivity determinants of arylsulfonamide $Na_V1.7$ inhibitors have been well studied. Mutation of key non-conserved residues proximal to the binding pocket observed in the **GX-936** co-structure has been shown to alter $Na_V1.7$ potency consistent with selectivity patterns observed for other $Na_V$ family members (*Ahuja et al., 2015*; *Figure 4—figure supplement 3*). Typically, arylsulfonamides have shown high levels of selectivity against $Na_V1.5$, $Na_V1.1$, and $Na_V1.4$, but lower selectivity against $Na_V1.2$ and $Na_V1.6$. In comparison, several acylsulfonamides have been characterized with very high selectivity over $Na_V1.2$ and $Na_V1.6$ (*Bankar et al., 2018*). Interestingly, residues forming the contours of the acylsulfonamide binding pocket show high sequence homology across $Na_V$ isoforms, suggesting that selectivity for this class is likely an allosteric phenomenon. Nonetheless, the disparate binding pockets for these two inhibitor classes provide a clear rationale for their historically divergent patterns of selectivity.

Our structure of **GDC-0310** provides a structural rationale for the relative slow dissociation kinetics of acylsulfonamide inhibitors compared to arylsulfonamides (*Bankar et al., 2018*). Specifically, the acylsulfonamide pocket is buried more deeply into the plasma membrane than the arylsulfonamide pocket, and may only be available through a membrane-access pathway that first involves partitioning of the molecule into the hydrophobic core of the membrane. In comparison, the arylsulfonamide pocket is more open to the outer membrane leaflet–aqueous interface, potentially offering access or egress directly from (or to) the solvent compartment. It is increasingly recognized that slow dissociation kinetics may be associated with membrane-access pathways for small molecules binding to transmembrane proteins (*Payandeh and Volgraf, 2021*; *Mason et al., 1991*; *Coleman et al., 1996*; *Anderson et al., 1994*; *Masureel et al., 2018*; *Austin et al., 2003*; *Rhodes et al., 1992*; *Sykes et al., 2014*; *Dickson et al., 2016*).

In addition to offering valuable information for retrospective analysis of historical $Na_V1.7$ VSD4 domain inhibitors, the structure of **GDC-0310** inspired the development of a class of structurally differentiated inhibitors that trap a novel conformation of VSD4. These hybrid molecules bridge both the S2/S3 and S3/S4 helical gaps, with the anionic moiety situated centrally, interacting with the arginine gating charge network within the core of the domain. We have demonstrated that potent hybrid inhibitors can be designed with fewer membrane-associated elements than prior generation

Na$_V$1.7 inhibitors, and anticipate that this will offer opportunities to develop molecules with improved lipophilic ligand efficiency and/or differentiated pharmacokinetic profiles. Hybrid inhibitors can also access selectivity determinants from both the S2/S3 and S3/S4 pockets simultaneously, offering a tantalizing opportunity to develop exquisite selectivity over other Na$_V$ isoforms. Although selectivity was not a key endpoint of our proof-of-concept studies described here, hybrid molecules **2–5** are selective for Na$_V$1.7 over the cardiac channel Na$_V$1.5. Further discussion of selectivity patterns in the hybrid series is included in *Supplementary file 1B*.

Beyond the scope of Na$_V$ channel inhibitors, our experiences may also offer lessons translatable to small molecule modulators of other transmembrane proteins. Specifically, it is noteworthy to observe that both aryl- and acylsulfonamide inhibitors have access to inducible pockets that are not observable in the parent small molecule co-structures. Designing small molecules to 'push' on a structural protein element is a common strategy in medicinal chemistry, but it is often challenging to predict when such efforts will afford new, well-defined binding pockets. As such, these designs are often met with failure and quickly abandoned. However, some scenarios exist in which an inducible pocket is more likely to be found. In these cases, medicinal chemists are more often emboldened to continue experimenting with designs that push into new areas of a protein binding pocket, even when met with initial failure. For example, type II kinase inhibitors can be rationally designed starting from a type I inhibitor structure by appending groups that push into the hydrophobic back pocket (*Zhao et al., 2014*). This pocket is not visible from the type I inhibitor structure, but knowledge of the protein dynamics associated with the DFG-in/DFG-out transition provides a rationale for exploring these designs. We speculate that transmembrane proteins, particularly intrinsically dynamic ones including ion channels, transporters, and GPCRs, may commonly present privileged structural motifs that are rich with opportunities for the discovery of novel druggable pockets.

In summary, we have developed a robust pipeline for high-resolution structural determination of small-molecules bound to Na$_V$1.7-VSD4 suitable for use in iterative structure-based drug design campaigns. We also report the first co-structure of an acylsulfonamide inhibitor bound to the VSD4 domain of Na$_V$1.7, revealing an unexpected and unique binding mode. This finding provides clear rationale for previously unexplainable divergence in the in vitro pharmacological behavior between inhibitor classes, and inspired the development of hybrid inhibitors that engages VSD4 in a novel conformation. Our findings highlight the power of cryo-EM as an enabling drug discovery technology and offer insights into ion channel structural dynamics that are potentially applicable to related targets and other target classes.

## Materials and methods
### Generation of VSD4-Na$_V$1.7-Na$_V$Pas channel constructs

VSD4-Na$_V$1.7-Na$_V$Pas chimeric constructs were used as described previously (*Clairfeuille, 2019*). In brief, optimized coding DNA for Na$_V$1.7 VSD4-Na$_V$PaS chimeras with N-terminal tandem StrepII and FLAG tag was cloned into a pRK vector with CMV promoter. HEK293 cells in suspension were cultured in SMM 293T-I medium under 5% CO$_2$ at 37°C and transfected using PEI when the cell density reached $4 \times 10^6$ cells per mL. Transfected cells were harvested 48 hr after transfection. The Dc1a toxin coding DNA from *Diguetia canities* was cloned into a modified pAcGP67A vector downstream of the polyhedron promoter and an N-terminal 6x HIS tag. Recombinant baculovirus was generated using the Baculogold system (BD Biosciences) and *Trichoplusia ni* cells were infected for protein production. The supernatant was harvested 48 hr post-infection.

### Protein expression and purification

In total, 150 g of cell pellet was resuspended in 500 mL of 25 mM HEPES pH 7.5, 200 mM NaCl, 1 ug/mL benzonase, 1 mM PMSF, and Roche protease inhibitor tablets. Cells were lysed by dounce homogenization. Proteins were solubilized by addition of 2% GDN (Avanti) with 0.3% cholesteryl hemisuccinate (Avanti) for 2 hr at 4°C. Cell debris was separated by ultracentrifugation at 40,000 rpm at 4°C. Affinity purification using anti-Flag resin was performed by batch binding for 1 hr at 4°C. The resin was washed with 5CV Purification Buffer (25 mM HEPES pH 7.5, 200 mM NaCl, 0.01% GDN). Another 5CV wash was performed using Purification Buffer supplemented with 5 mM ATP and 10 mM MgCl$_2$. Proteins were eluted in 6CV Purification Buffer with 300 ug/mL Flag peptide. Proteins were subjected

to another round of affinity purification using StrepTactin resin (IBA). Proteins were eluted and concentrated to ~5 mg/mL. For the compound **2** containing sample, instead of nanodisc reconstitution VSD4-Na$_V$1.7-Na$_V$Pas was incubated overnight with Dc1a toxin at a 2:1 molar ration of toxin:VSD4-Na$_V$1.7-Na$_V$Pas. The eluted sample was concentrated to 100 uL and separated on Superose 6 3.2/300.

For nanodisc reconstitution used for the samples containing **GNE-3565, GDC-0310,** and compound **4**, a 200-molar excess of lipid mix (3POPC:1POPE:1POPG resuspended in 50 mM HEPES pH7.5, 100 mM NaCl, 5 mM MgCl$_2$, 1% CHAPS) was added to detergent-solubilized protein and incubated on ice for 30 min. A four-molar excess of scaffold protein (MSP1E3D1, Sigma) was added to the protein-lipid mix and incubated on ice for another 30 min. To remove detergent, BioBeads (Bio-Rad) were added to 0.25 mg/mL and incubated overnight at 4°C. To remove the empty nanodiscs, the sample was subjected to a round of affinity purification using Strep-Tactin (IBA). The eluted sample was concentrated to 100 uL and separated on Superose 6 3.2/300. Peak fractions were combined and split into four samples. Then, 50 uM of the small molecule of interest was added to each sample and incubated at 22°C for 10 min. Samples were crosslinked with 0.05% glutaraldehyde (Electron Microscopy Sciences) then quenched with 1 M Tris pH 7.0. Samples at 2 mg/mL were used for grid freezing.

## Cryo-EM sample preparation and data acquisition

Cryo-EM grids for the small molecule VSD4-Na$_V$1.7-Na$_V$Pas chimeric complexes shown in this study were prepared as follows: for the **GDC-0310** and compound **4** VSD4-Na$_V$1.7-Na$_V$Pas complexes, holey carbon grids (Ultrafoil 25 nM Au R 0.6/1 300 mesh; Quantifoil) were incubated with a thiol reactive, self-assembling reaction mixture of 4 mM monothiolalkane(C11)PEG6-OH (11-mercaptoundecyl) hexaethyleneglycol (SPT-0011P6, SensoPath Technologies, Inc, Bozeman, MT). Grids were incubated with this self-assembled monolayer (SAM) solution for 24 hr and afterward rinsed with EtOH. 3 µL of the sample was applied to the grid and blotted with Vitrobot Mark IV (Thermo Fisher) using 3.5 s blotting time with 100% humidity and plunge-frozen in liquid ethane cooled by liquid nitrogen. The **GNE-3565** VSD4-Na$_V$1.7-Na$_V$Pas complex was, similarly as described above, applied to a holey carbon grid (Ultrafoil 25 nM Au R 1.2/1,3 300 mesh; Quantifoil) pretreated with SAM solution. The grid was blotted single-sided with a Leica EM GP (Leica) using 3 s blotting time with 100% humidity and plunge-frozen in liquid ethane cooled by liquid nitrogen. For compound **2** VSD4-Na$_V$1.7-Na$_V$Pas-DC1a complex, holey carbon grids (Ultrafoil 25 nM Au R 2/2 200 mesh; Quantifoil) were glow-discharged for 10 s using the Solarus plasma cleaner (Gatan). 3 µL of the sample was applied to the grid and blotted with Vitrobot Mark IV (Thermo Fisher) using 2.5 s blotting time with 100% humidity and plunge-frozen in liquid ethane cooled by liquid nitrogen.

Movie stacks for compound **2** VSD4-Na$_V$1.7-Na$_V$Pas-DC1a complex were collected using SerialEM (*Mastronarde, 2005*) on a Titan Krios operated at 300 keV with bioquantum energy filter equipped with a K2 Summit direct electron detector camera (Gatan). Images were recorded at ×165,000 magnification corresponding to 0.824 Å per pixel using a 20 eV energy slit. Each image stack contains 50 frames recorded every 0.2 s for an accumulated dose of ~50 e Å$^{-2}$ and a total exposure time of 10 s. Images were recorded with a set defocus range of 0.5–1.5 µm.

Movie stacks for **GNE-3565, GDC-0310,** and compound **4** VSD4-Na$_V$1.7-Na$_V$Pas complexes were collected using SerialEM on a Titan Krios G3i (Thermo Fisher Scientific, Waltham, MA) operated at 300 keV with bioquantum energy filter equipped with a K3 Summit direct electron detector camera (Gatan Inc, Pleasanton, CA). Images were recorded in EFTEM mode at ×105,000 magnification corresponding to 0.838 Å per pixel, using a 20 eV energy slit. Each image stack contains 60 frames recorded every 0.05 s for an accumulated dose of ~60 e Å$^{-2}$ and a total exposure time of 3 s. Images were recorded with a set defocus range of 0.5–1.5 µm.

## Cryo-EM data processing

Cryo-EM data were processed using a combination of the RELION (*Scheres, 2012*) and cisTEM (*Grant et al., 2018*) software packages, similarly as described previously (*Kschonsak et al., 2022*) and as illustrated in *Figure 2—figure supplement 1* and *Figure 3—figure supplement 1*.

Movies were corrected for frame motion using the MotionCor2 (*Zheng et al., 2017*) implementation in RELION, and contrast-transfer function parameters were fit using the 30–4.5 Å band of the spectrum with CTFFIND-4 (*Rohou and Grigorieff, 2015*). CTF-fitted images were filtered on the basis of the detected fit resolution better than 6–10 Å. Particles were picked by template-matching with

Gautomatch using a 30 Å low-pass filtered apo VSD4-Na$_V$1.7-Na$_V$Pas reference structure. Particles were sorted during RELION 2D classification, and selected particles were imported into cisTEM for 3D refinements. 3D reconstructions were obtained after auto-refine and manual refinements with a mask around the channel (excluding the detergent micelle) and by applying low-pass filter outside the mask (filter resolution 20 Å) and a score threshold of 0.10–0.30, so that only the best-scoring 10–30% of particle images would be included in the 3D reconstruction at each cycle. The weight outside of the mask was set to 0.8. No data beyond 3.0 Å for compound **4**, 3.4 Å for **GDC-0310**, 3.7 Å **GNE-3565,** and 4.0 Å for compound **2** were used in the refinements. Phenix ResolveCryoEM (*Terwilliger et al., 2020*) density modification was applied to each of the reconstructions to obtained the final map used for model building. Local resolution was determined in cisTEM using a reimplementation of the blocres algorithm.

## Model building and structure analysis

The previously determined VSD4-Na$_V$1.7-Na$_V$Pas model (PDB: 6NT3; *Clairfeuille, 2019*) was fit as a rigid body into the cryo-EM map. After manual adjustments, multiple rounds of real space refinement using the phenix.real_space_refinement tool (*Afonine et al., 2018b*) were used to correct structural differences between the initial model and the map. The small molecule coordinates were generated with eLBOW (*Moriarty et al., 2009*), placed and manually adjusted in Coot (*Emsley et al., 2010*), energy minimized with MOE (Chemical Computing Group ULC), and refined with real space refinements in Phenix (*Afonine et al., 2018b*). The model was validated using phenix.validation_cryoem (*Afonine et al., 2018a*) with built-in MolProbity scoring (*Williams et al., 2018*). Figures were made using PyMOL (The PyMOL Molecular Graphics System, v.2.07 Schrödinger, LLC), UCSF Chimera (*Pettersen et al., 2004*), and UCSF ChimeraX (*Goddard et al., 2018*).

## Chemical synthesis

The synthesis of **GNE-3565**, **GDC-0310**, and compound **1** has been previously reported (*Safina et al., 2021*; *Sun et al., 2019*; *Sutherlin, 2021*). The synthesis and characterization of compounds **2–5** are described in the supporting information.

## Na$_V$1.7 biophysical and pharmacological characterization of Na$_v$1.7 channel using Syncropatch electrophysiology

cDNA for Na$_V$1.7 (NM_002977) were stably expressed in Chinese Hamster Ovary (CHO) cells. Sodium currents were measured in the whole-cell configuration using Syncropatch 384PE (NanIon Technologies, Germany). 1NPC–384 chips with custom medium resistance and single hole mode were used. Internal solution consisted of (in mM) 110 CsCl, 10 CsCl, 20 EGTA, and 10 HEPES (pH adjusted to 7.2); and external solution contains (in mM) 60 NMDG, 80 NaCl, 4 KCl, 1 MgCl$_2$, 2 CaCl$_2$, 2 D-glucose monohydrate, and 10 HEPES (pH adjusted to 7.4 with NaOH). To assess voltage-dependent activation of Na$_V$1.7, currents were elicited by 20 ms test pulses (−80 to 20 mV in 5 mV increments) from a holding potential at –120 mV. To determine voltage-dependent inactivation, peak currents at 0 mV were obtained after 500 ms conditioning pre-pulses varying from –120–15 mV.

After system flushing, testing compounds were dissolved in external solution containing 0.1% Pluronic F-127. 10 μL cells were added to the chip from a cell hotel, and a negative pressure of −50 mBar was applied to form a seal. Following treatment with seal enhancer solution and wash-off with external solution, negative pressure of −250 mbar was applied for 1 s to achieve the whole-cell configuration, followed by three washing steps in external solution. 20 μL of compounds were added to 40 μL in each well (1:3 dilution of compounds), and after mixing, 20 μL was removed so the volume was retained at 40 uL. After approximately 13 min recording, 20 μL/well of 2 uM TTX was added to achieve full block.

For pharmacological characterization, a holding potential of –50 mV was applied during the whole experiment. A depolarizing step was applied to –10 mV for 10 ms, followed by a hyperpolarization step to –150 mV for 20 ms to allow channel recovery from inactivation. A second depolarizing step was applied from –150 mV to –10 mV for 10 ms, where currents were measured to derive blocking effects of compounds. Inhibition was determined based on 7.5 min of compound incubation.

For all recordings, currents were sampled at 10 kHz and filtered with Bessel filter. Seal resistance (Rseal) was calculated using built-in protocols; and series resistance was compensated at 80%.

Several parameters were applied for quality control, including: cell catching (>10 MU), seal resistance (>500 MΩ), series resistances (<10 MΩ), and baseline current amplitude (>500 pA and <5 nA). Data were further examined by manual inspection.

Biophysical and pharmacological characterizations, including voltage-dependent activation, voltage-dependent inactivation, pharmacological protocol, a sample trace, IC50 values, and individual dose responses are shown in *Figure 4—figure supplement 1*.

## Acknowledgements

We are grateful to BMR and RMG for construct generation and protein scale-up. We are thankful to Cameron Noland for help with the biochemistry of the VSD4-Na$_V$1.7-Na$_V$Pas channel system. We thank Tommy Lai and Wenfeng Liu for chemistry support.

## Additional information

### Competing interests

Marc Kschonsak, Christine C Jao, Christopher P Arthur, Alexis L Rohou, Philippe Bergeron, Daniel F Ortwine, Steven J McKerrall, David H Hackos, Lunbin Deng, Jun Chen, Tianbo Li, Peter S Dragovich, Matthew Volgraf, Matthew R Wright, Jian Payandeh, Claudio Ciferri, John C Tellis: are Genentech employees.

### Funding

| Funder | Grant reference number | Author |
| --- | --- | --- |
| Genentech | | Marc Kschonsak |

The funders had no role in study design, data collection and interpretation, or the decision to submit the work for publication.

### Author contributions

Marc Kschonsak, Data curation, Software, Formal analysis, Validation, Investigation, Visualization, Methodology, Writing – original draft, Writing – review and editing; Christine C Jao, Data curation, Formal analysis, Validation, Visualization, Methodology; Christopher P Arthur, Formal analysis, Methodology; Alexis L Rohou, Philippe Bergeron, Methodology; Daniel F Ortwine, Steven J McKerrall, David H Hackos, Lunbin Deng, Jun Chen, Matthew Volgraf, Formal analysis; Tianbo Li, Data curation, Formal analysis; Peter S Dragovich, Matthew R Wright, Conceptualization; Jian Payandeh, Conceptualization, Supervision, Investigation, Writing – original draft, Writing – review and editing; Claudio Ciferri, Conceptualization, Supervision, Funding acquisition, Validation, Investigation, Visualization, Methodology, Writing – original draft, Project administration, Writing – review and editing; John C Tellis, Conceptualization, Data curation, Formal analysis, Supervision, Investigation, Visualization, Writing – original draft, Writing – review and editing

### Author ORCIDs
Marc Kschonsak ⓘ http://orcid.org/0000-0002-4392-1614
Claudio Ciferri ⓘ http://orcid.org/0000-0002-0804-2411
John C Tellis ⓘ http://orcid.org/0000-0003-1675-9639

### Decision letter and Author response
Decision letter https://doi.org/10.7554/eLife.84151.sa1
Author response https://doi.org/10.7554/eLife.84151.sa2

## Additional files

### Supplementary files
• Supplementary file 1. Experimental details for Cryo-EM, compound synthesis and characterization, and NaV subtype selectivity for selected Na$_V$1.7 inhibitors. (A) Cryogenic electron microscopy (cryo-

EM) data collection, refinement and validation statistics. (B) Na$_V$ subtype selectivity for selected hybrid compounds.

- MDAR checklist

## Data availability

All data generated or analyzed during this study are included in the manuscript and supporting files. The NaV1.7-NaVPas/GNE-3565 coordinates and cryo-EM maps were deposited in the PDB entry ID 8F0R and EMDB entry ID EMD-28778, respectively. The NaV1.7-NaVPas/GDC-0310 coordinates and cryo-EM maps were deposited in the PDB entry ID 8F0Q and EMDB entry ID EMD-28777, respectively. The NaV1.7-NaVPas bound to the hybrid molecule 2 (GNE-9296) coordinates and cryo-EM maps were deposited in the PDB entry ID 8F0S and EMDB entry ID EMD-28779, respectively. The NaV1.7-NaVPas bound to the hybrid molecule 4 (GNE-1305) coordinates and cryo-EM maps were deposited in the PDB entry ID 8F0P and EMDB entry ID EMD-28776, respectively.

The following datasets were generated:

| Author(s) | Year | Dataset title | Dataset URL | Database and Identifier |
|---|---|---|---|---|
| Kschonsak M, Jao CC, Arthur CP, Rohou AL, Bergeron P, Otwine D, McKerall SJ, Hackos DH, Deng L, Chen J, Sutherlin D, Dragovich PS, Volgraf M, Wright MR, Payandeth J, Ciferri C, Tellis JC | 2023 | Structure of VSD4-NaV1.7-NaVPas channel chimera bound to the arylsulfonamide inhibitor GNE-3565 | https://www.rcsb.org/structure/8F0R | RCSB Protein Data Bank, 8F0R |
| Kschonsak M, Jao CC, Arthur CP, Rohou AL, Bergeron P, Otwine D, McKerall SJ, Hackos DH, Deng L, Chen J, Sutherlin D, Dragovich PS, Volgraf M, Wright MR, Payandeh J, Ciferri C, Tellis JC | 2023 | Structure of VSD4-NaV1.7-NaVPas channel chimera bound to the arylsulfonamide inhibitor GNE-3565 | https://www.ebi.ac.uk/emdb/EMD-28778 | Electron Microscopy Data Bank, EMD-28778 |
| Kschonsak M, Jao CC, Arthur CP, Rohou AL, Bergeron P, Ortwine D, McKerall SJ, Hackos DH, Deng L, Chen J, Sutherlin D, Dragovich PS, Volgraf M, Wright MR, Payandeh J, Ciferri C, Tellis JC | 2023 | Structure of VSD4-NaV1.7-NaVPas channel chimera bound to the acylsulfonamide inhibitor GDC-0310 | https://www.rcsb.org/structure/8F0Q | RCSB Protein Data Bank, 8F0Q |
| Kschonsak M, Jao CC, Arthur CP, Rohou AL, Bergeron P, Ortwine D, McKerall SJ, Hackos DH, Deng L, Chen J, Sutherlin D, Dragovich PS, Volgraf M, Wright MR, Payandeh J, Ciferri C, Tellis JC | 2023 | Structure of VSD4-NaV1.7-NaVPas channel chimera bound to the acylsulfonamide inhibitor GDC-0310 | https://www.ebi.ac.uk/emdb/EMD-28777 | Electron Microscopy Data Bank, EMD-28777 |

*Continued on next page*

*Continued*

| Author(s) | Year | Dataset title | Dataset URL | Database and Identifier |
|---|---|---|---|---|
| Kschonsak M, Jao CC, Arthur CP, Rohou AL, Bergeron P, Ortwine D, McKerall SJ, Hackos DH, Deng L, Chen J, Sutherlin D, Dragovich PS, Volgraf M, Wright MR, Payandeh J, Ciferri C, Tellis JC | 2023 | Structure of VSD4-NaV1.7-NaVPas channel chimera bound to the hybrid inhibitor GNE-9296 | https://www.rcsb.org/structure/8F0S | RCSB Protein Data Bank, 8F0S |
| Kschonsak M, Jao CC, Arthur CP, Rohou AL, Bergeron P, Ortwine D, McKerall SJ, Hackos DH, Deng L, Chen J, Sutherlin D, Dragovich PS, Volgraf M, Wright MR, Payandeh J, Ciferri C, Tellis JC | 2023 | Structure of VSD4-NaV1.7-NaVPas channel chimera bound to the hybrid inhibitor GNE-9296 | https://www.ebi.ac.uk/emdb/EMD-28779 | Electron Microscopy Data Bank, EMD-28779 |
| Kschonsak M, Jao CC, Arthur CP, Rohou AL, Bergeron P, Ortwine D, McKerall SJ, Hackos DH, Deng L, Chen J, Sutherlin D, Dragovich PS, Volgraf M, Wright MR, Payandeh J, Ciferri C, Tellis JC | 2023 | Structure of VSD4-NaV1.7-NaVPas channel chimera bound to the hybrid inhibitor GNE-1305 | https://www.rcsb.org/structure/8F0P | RCSB Protein Data Bank, 8F0P |
| Kschonsak M, Jao CC, Arthur CP, Rohou AL, Bergeron P, Ortwine D, McKerall SJ, Hackos DH, Deng L, Chen J, Sutherlin D, Dragovich PS, Volgraf M, Wright MR, Payandeh J, Ciferri C, Tellis JC | 2023 | Structure of VSD4-NaV1.7-NaVPas channel chimera bound to the hybrid inhibitor GNE-1305 | https://www.ebi.ac.uk/emdb/EMD-28776 | Electron Microscopy Data Bank, EMD-28776 |

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
