## [Editor Report]

This fundamental study describes the structure-based design of novel hybrid inhibitors targeting a human sodium channel which is a pain target. Exceptionally strong evidence for key claims was produced with a structural biological pipeline for iterative structural determination of drugs complexed with an engineered sodium channel. This work is expected to be of interest to biophysicists, drug developers, neurobiologists and pain researchers.

---

## [Decision Letter]

**Decision letter after peer review:**

Thank you for submitting your article "CryoEM reveals unprecedented binding site for NaV1.7 inhibitors enabling rational design of potent hybrid inhibitors" for consideration by *eLife*. Your article has been reviewed by 3 peer reviewers, and the evaluation has been overseen by a Reviewing Editor and Richard Aldrich as the Senior Editor. The following individuals involved in the review of your submission have agreed to reveal their identity: Vladimir Yarov-Yarovoy (Reviewer #1); Michael A Cianfrocco (Reviewer #2); Hiro Furukawa (Reviewer #3).

Essential revisions:

1) Please show electrophysiological data along with statistical analyses.

2) Clarify what the advances of the structural biological pipeline are. Based on the description of the pipeline given, reviewers felt the advance was in protein engineering, rather than the implementation of cryo-EM.

3) Address each reviewer's suggestions with changes to the manuscript or explain why manuscript changes are not warranted in rebuttal.

*Reviewer #1 (Recommendations for the authors):*

1. Add main text and Supplementary figures showing Syncropatch electrophysiology potency data for all compounds presented in the manuscript.

2. Include electrophysiology selectivity data for human Nav1.7 versus other human Nav channels for at least one of the novel hybrid inhibitors. These data will be useful to compare the selectivity profiles of the novel hybrid inhibitors to the selectivity profiles of arylsulfonamides and acylsulfonamides.

3. Discussion of Nav channel subtype selectivity determinants highlighting residues Y1537 and W1538 would benefit from corresponding multiple sequence alignment of Nav channels in a Supplementary Figure.

4. Add supplementary figures showing hybrid compounds 4 and 5 in complex with VSD4 NaV1.7-NaVPas channel using a layout similar to main text Figures 2 and 3.

5. Clarify in the main text and figure legends that compound 2 is GNE-9296.

6. Clarify in the main text and figure legends that compound 4 is GNE-1305.

7. Explain why DC1a peptide toxin was specifically used to obtain the cryoEM structure of compound 2 in complex with VSD4 NaV1.7-NaVPas channel.

8. Update Supplementary Figure 4 to reflect the Supplementary Figure 4 legend.

9. Add Supplementary Figure 5 to reflect the Supplementary Figure 5 legend.

10. Main text line 205 – add "s" to "suggest" to read "our structure suggests".

*Reviewer #2 (Recommendations for the authors):*

– One source of weakness can be identified in lines 43-45, where the authors emphasize an established iterative system of structure determination, which has been instrumental to their success in drug development. As such, if the authors want to play up their pipeline, they should elaborate on details of the pipeline: how long did it take for each step, how are class averages selected, how long does it take to go from cryo-EM data collection to final reconstruction and atomic model? At the moment, it is difficult to tell how innovative the pipeline is compared to conventional cryo-EM structure determination. Otherwise, if the authors do not want to expound on this, they can omit their comments on a pipeline.

– The density-modified reconstructions show sharp edges that are non-physiological. The authors need to remedy this to show density that reflects protein structure (i.e., smooth). One approach would include creating composite maps with appropriate local resolutions and B-factors to prevent over-sharpening artifacts.

– Missing legend for Figure 1F-G.

– There are two Supplementary Figure 1 figures. This makes the supplementary figure legend numbering incorrect.

---

## [Author Response]

Reviewer #1 (Recommendations for the authors):1. Add main text and Supplementary figures showing Syncropatch electrophysiology potency data for all compounds presented in the manuscript.

Electrophysiology data for all compounds has been added in Figure4—figure supplement 1.

2. Include electrophysiology selectivity data for human Nav1.7 versus other human Nav channels for at least one of the novel hybrid inhibitors. These data will be useful to compare the selectivity profiles of the novel hybrid inhibitors to the selectivity profiles of arylsulfonamides and acylsulfonamides.

We agree with this reviewer that subtype selectivity is an important characteristic of Na_V_ channel inhibitors. We have included selectivity data for various molecules against Na_V_1.5 and Na_V_1.6 in Supplementary File 1B. This table is also accompanied by a brief discussion. In addition, the following line has been added to the discussion in the main text:

“Although selectivity was not a key endpoint of the proof-of-concept studies described here, hybrid molecules 2-5 are selective for Na_V_1.7 over cardiac channel Na_V_1.5. Further discussion of selectivity patterns in the hybrid series is included in the Supplementary Information (Supplementary File 1B).”

3. Discussion of Nav channel subtype selectivity determinants highlighting residues Y1537 and W1538 would benefit from corresponding multiple sequence alignment of Nav channels in a Supplementary Figure.

We thank the reviewer for this suggestion and we have now added a VSD4 sequence alignment to Figure4—figure supplement 3. We also added further discussion of selectivity pattern in the hybrid series in Supplementary File 1B.

4. Add supplementary figures showing hybrid compounds 4 and 5 in complex with VSD4 NaV1.7-NaVPas channel using a layout similar to main text Figures 2 and 3.

We do present the structure of compound 4 (G-1305) in Figure 4G-I. We did not solve the structure of compound 5 because of its lower potency. We believe that adding the structure of Nav bound to 5 would not add significantly to this manuscript.

5. Clarify in the main text and figure legends that compound 2 is GNE-9296.

We have changed this throughout the text and figures. GNE-9296 is added as an alias for compound 2.

6. Clarify in the main text and figure legends that compound 4 is GNE-1305.

We have changed this throughout the text and figures. GNE-1305 is added as an alia of compound 4.

7. Explain why DC1a peptide toxin was specifically used to obtain the cryoEM structure of compound 2 in complex with VSD4 NaV1.7-NaVPas channel.

We have worked on optimizing the Nav sample during the last several years, exploring different constructs and purification protocols. Originally, we were performing cryoEM structure determination on samples purified in detergent and further stabilized with DC1a. We later moved to nanodisc (in the absence of DC1a) because most of the compound bound samples, purified in nanodisc, refined to better resolution. GNE-9296 represented an exception to this general trend and the structure of Nav bound to this compound, when purified in detegerents in the presence of DC1a, went to the highest resolution. This is the reason why we decided to include this specific structure in the manuscript.

8. Update Supplementary Figure 4 to reflect the Supplementary Figure 4 legend.

We apologize for this mismatch; this is now fixed.

9. Add Supplementary Figure 5 to reflect the Supplementary Figure 5 legend.

We apologize for this mismatch; this is now fixed.

10. Main text line 205 – add "s" to "suggest" to read "our structure suggests".

We apologize for this typo; this is now fixed.

Reviewer #2 (Recommendations for the authors):– One source of weakness can be identified in lines 43-45, where the authors emphasize an established iterative system of structure determination, which has been instrumental to their success in drug development. As such, if the authors want to play up their pipeline, they should elaborate on details of the pipeline: how long did it take for each step, how are class averages selected, how long does it take to go from cryo-EM data collection to final reconstruction and atomic model? At the moment, it is difficult to tell how innovative the pipeline is compared to conventional cryo-EM structure determination. Otherwise, if the authors do not want to expound on this, they can omit their comments on a pipeline.

We thank this Reviewer for bringing up this specific point. We were interested in performing Structural Based Drug Design (SBDD) on this channel for the last 10 years, initially through X-ray crystallography and then, starting in 2017, by cryoEM. In order to perform real time SBDD, it is imperative that the cycle of structure determination matches the cycle of design and synthesis of novel compounds. Initially, the time comprising protein purification, grid preparation, data collection and structure determination was approximately 1.5 months/compound. We worked on every single step to diminish this timeline and bring this to 2 weeks. This included having a more robust sample (the optimum was by utilizing nanodiscs), optimize grid freezing and data collection (to maximize number of particles and orientation for structure determination), and finally streamline data processing (based on our improved experience with this sample). We do realize that in the meantime the cryoEM field has significantly improved making these optimizations very common for challenging samples. That said, reported small molecule cryoEM structures are generally retrospective, including the ones showcasing cutting edge technology (i.e. Terrett et al., 2021). This manuscript represents, to our knowledge, one of the very first examples of real-time SBDD and utilization of cryoEM to design a new class of compounds with improved properties and potential for clinical studies. We have changed the text to:

“To advance inhibitor design targeting the NaV1.7 channel, we established an iterative system to routinely obtain high-resolution ligand-bound NaV1.7 structures using cryogenic electron microscopy (cryo-EM).”

Citation:

Tetrahydrofuran-Based Transient Receptor Potential Ankyrin 1 (TRPA1) Antagonists: Ligand-Based Discovery, Activity in a Rodent Asthma Model, and Mechanism-of-Action via Cryogenic Electron Microscopy.

Terrett JA, Chen H, Shore DG, Villemure E, Larouche-Gauthier R, Déry M, Beaumier F, Constantineau-Forget L, Grand-Maître C, Lépissier L, Ciblat S, Sturino C, Chen Y, Hu B, Lu A, Wang Y, Cridland AP, Ward SI, Hackos DH, Reese RM, Shields SD, Chen J, Balestrini A, Riol-Blanco L, Lee WP, Liu J, Suto E, Wu X, Zhang J, Ly JQ, La H, Johnson K, Baumgardner M, Chou KJ, Rohou A, Rougé L, Safina BS, Magnuson S, Volgraf M. J Med Chem. 2021 Apr 8;64(7):3843-3869.

– The density-modified reconstructions show sharp edges that are non-physiological. The authors need to remedy this to show density that reflects protein structure (i.e., smooth). One approach would include creating composite maps with appropriate local resolutions and B-factors to prevent over-sharpening artifacts.

We thank the reviewer for this suggestion. We have also deposited the half-maps and non-modified full maps that show the smooth nature of the protein.

– Missing legend for Figure 1F-G.

We thank the reviewer for pointing out this error. We have now corrected the Figure and the Figure legend information.

– There are two Supplementary Figure 1 figures. This makes the supplementary figure legend numbering incorrect.

We thank the reviewer for pointing out this error. We have corrected the numbering of the Supplementary Figure legends.